

# An updated version of a gap-free monthly mean zonal mean ozone database

Birgit Hassler[1,2], Stefanie Kremser[1], Greg E. Bodeker[1], Jared Lewis[1], Kage Nesbit[1], Sean M. Davis[3,4], Martyn P. Chipperfield[5,6], Sandip S. Dhomse[5,6], and Martin Dameris[2]

[1]Bodeker Scientific, 42 Russell Street, Alexandra, 9320, New Zealand
[2]Deutsches Zentrum für Luft- und Raumfahrt (DLR), Institut für Physik der Atmosphäre, Oberpfaffenhofen, Germany
[3]Cooperative Institute for Research in Environmental Sciences, University of Colorado, Boulder, CO, USA
[4]NOAA Earth System Research Laboratory, Boulder, CO, USA
[5]School of Earth and Environment, University of Leeds, Leeds, UK
[6]National Centre for Earth Observation, University of Leeds, Leeds, UK

**Correspondence:** Birgit Hassler (birgit.hassler@dlr.de)

**Abstract.** An updated and improved version of a global, vertically resolved, monthly mean zonal mean ozone database has been calculated – hereafter referred to as the BSVertOzone database. Like its predecessor, it combines measurements from several satellite-based instruments and ozone profile measurements from the global ozonesonde network. Monthly mean zonal mean ozone concentrations in mixing ratio and number density are provided in 5°latitude bins, spanning 70 altitude levels (1 to 70km), or 70 pressure levels that are approximately 1km apart (878.4hPa to 0.046hPa). Different data sets or "Tiers" are provided: "Tier 0" is based only on the available measurements and therefore does not completely cover the whole globe or the full vertical range uniformly; the "Tier 0.5" monthly mean zonal means are calculated as a filled version of the Tier 0 database where missing monthly mean zonal mean values are estimated from correlations against a total column ozone database. The Tier 0.5 data set includes the full range of measurement variability and is created as an intermediate step for the calculation of the "Tier 1" data where a least squares regression model is used to attribute variability to various known forcing factors for ozone. Regression model fit coefficients are expanded in Fourier series and Legendre polynomials (to account for seasonality and latitudinal structure, respectively). Four different combinations of contributions from selected regression model basis functions result in four different "Tier 1" data sets that can be used for comparisons with chemistry-climate model simulations that do not exhibit the same unforced variability as reality (unless they are nudged towards reanalyses). Compared to previous versions of the database, this update includes additional satellite data sources and ozonesonde measurements to extend the database period to 2016. Additional improvements over the previous version of the database include: (i) Adjustments of measurements to account for biases and drifts between different data sources (using a chemistry-transport model simulation as a transfer standard), (ii) a more objective way to determine the optimum number of Fourier and Legendre expansions for the basis function fit coefficients, and (iii) the derivation of methodological and measurement uncertainties on each database value are traced through all data modification steps. Comparisons with the ozone database from SWOOSH (Stratospheric Water and OzOne Satellite Homogenized data set) show good agreement in many regions of the globe. Minor differences are caused





by different bias adjustment procedures for the two databases. However, compared to SWOOSH, BSVertOzone additionally covers the troposphere. Version 1.0 of BSVertOzone is publicly available at http://doi.org/10.5281/zenodo.1217184.

## 1 Introduction

Ozone is a greenhouse gas, and changes in stratospheric ozone concentrations have an effect on surface climate. Ozone changes
can result in direct radiative forcing changes (e.g. Forster, 1999), changes in surface UV radiation (e.g. McKenzie et al., 2011), and can alter the natural modes of tropospheric climate variability (e.g. Thompson et al., 2011). An accurate representation of stratospheric ozone in climate models is therefore essential if they are to be used to project realistic future climate behavior (e.g. Nowack et al., 2015). While many climate models calculate stratospheric ozone distributions with a fully coupled stratospheric chemistry scheme, Eyring et al. (2016) reported that not all models participating in phase 6 of the Coupled Model Intercom-
parison Project (CMIP6) will have that ability, and therefore need to prescribe atmospheric ozone concentrations. This requires a long-term, latitudinally and vertically resolved ozone database.

Vertically resolved ozone databases are not only useful for prescribing ozone concentrations in climate models but can also be used for climate model evaluation and development. When chemistry-climate models (CCMs) are run with specified dynamics (e.g. wind and temperature fields from reanalyses), the resulting ozone distributions are as close to the real atmospheric
distribution of ozone as CCMs can be expected to simulate them. Comparisons with observation-based, vertically resolved ozone databases can reveal possible model deficiencies in simulating chemical and dynamical processes. Diagnosing the problems occurring in the specified dynamics model simulations can inform a process-oriented validation of a free running CCM, and thereby improve the quality of projections.

Randel and Wu (2007) pioneered the idea of combining ozone measurements from different observation platforms, using
statistical methods for gap filling to create a long-term, gap-free global database of monhtly mean zonal mean stratospheric ozone values that can be used to evaluate CCM simulations and investigate ozone variability and trends. An updated and extended version of this database was created in support of the CMIP5 simulations (Cionni et al., 2011). The updated database, built by combining CCM simulations and observations, covered not just the historical period (from 1850), but also extended into the future (to 2100). This database was used by CMIP5 climate models that did not simulate their own stratospheric ozone,
and it also covered the troposphere since tropospheric ozone also has a radiative effect on the climate system. A first database that covered the troposphere and the stratosphere, based solely on measurements and using statistical methods for data gap-filling, was introduced by Bodeker et al. (2013) (hereafter BDBP v1.1.0.6), and is the predecessor of the database described here.

In preparation for the World Meteorological Organization/United Nations Environment Programme (WMO/UNEP) Sci-
entific Assessment of Ozone Depletion 2014, the communities involved in making satellite-based and ground-based ozone measurements decided to intensify the preparation of ozone databases that consist of multiple data sources from different platforms for stratospheric ozone variability investigations and trend detection. Several ozone databases were created that combine ('merge') measurements from (i) the same type of instruments that were flown on different satellites (Frith et al., 2017; Wild



and Long, 2018), (ii) two different satellite instruments (Kyrölä et al., 2013; Bourassa et al., 2014), or (iii) several different satellite instruments (Froidevaux et al., 2015; Davis et al., 2016). All of these databases cover the stratosphere, but only parts of the troposphere, if at all. Systematic comparisons between the databases showed that the applied merging technique plays a role in the representation of realistic long-term stratospheric ozone trends and variability (Tummon et al., 2015), beside the choice

of data sources. Recognizing these sensitivities, a second generation of most of these databases has already been developed, with reprocessed data sources and improved merging techniques (e.g. Bourassa et al., 2018; Sofieva et al., 2017; Zawada et al., 2017). However, these updated databases still mostly cover not all levels of the troposphere.

When combining measurements from different data sources, providing realistic uncertainty estimates on every value of the final data product (either calculated from the different data sources or estimated using statistical methods) becomes more and

more complex. However, realistic estimates of uncertainties on every datum are necessary to be able to estimate resultant uncertainties on ozone trends calculated from those data. This is particularly important when seeking to detect the signal of possible ozone recovery from the effect of ozone depleting substances that is still small (e.g. Harris et al., 2015). Measurements from different satellite instruments can exhibit offsets and drifts between coinciding measurements or when compared with ground-based measurements (see Hubert et al., 2016). When measurements are adjusted to account for any offsets and drifts,

it introduces an additional source of uncertainty which must also be accounted for. Diligent propagation of uncertainties from the individual measurements to the final data product, e.g. a monthly mean zonal mean value, is therefore essential.

Here, BSVertOzone v1.0 (Bodeker Scientific Vertical Ozone, hereafter referred to as "BSVertOzone") is described, which is an update and further developed version of the BDBP (Binary Database of Profiles) v1.1.0.6 that is described in Bodeker et al. (2013) and, as in that earlier version, consists of monthly mean zonal mean ozone values between 1km and 70km (878.4hPa

to 0.046hPa). The database was assigned a new name and not just a new version number due to the many improvements compared to the earlier version, and the fact that the data sources for the database are not available in binary format anymore. The following major improvements over BDBP v1.1.0.6 have been made: (1) where updated versions of the ozone data sources were available, these were used (Sect. 2); (2) data from the Microwave Limb Sounder and recent ozonesonde measurements were used as additional data sources Section 2); (3) drifts and biases between the data sources are now quantified and corrected

for using a chemistry-transport model as a transfer standard (Sect. 3); and (4) measurement uncertainties and uncertainties from other sources (e.g. applied offset and bias corrections) are propagated through to the final monthly mean zonal mean values (Sect. 2.2 and Sect. 3). In addition, a method was applied to estimate missing monthly mean zonal mean values before the database was subjected to regression modeling to generate data sets that include the effects of subsets of ozone forcing factors. The pre-filling of the database is described in Sect. 4. This filled data set is our best estimate of the true vertically and

latitudinally resolved monthly mean zonal mean ozone field. However, it is not suitable for direct comparison with ozone fields simulated by CCMs since it contains significant unforced variability. Therefore, in much the same way as described in Bodeker et al. (2013), a regression model was applied but, rather than using it to both fill the database and conduct an attribution of the various factors affecting ozone, it is now used only to conduct the attribution (Sect. 4). The fact that the regression model is applied to a pre-filled data set, makes the regression model fits more robust than it was the case for the BDBP v1.1.0.6 database.



A detailed description and validation of these global gap-free data sets is also presented (Sect. 5 and Sect. 6), followed by a conclusion (Sect. 7).

## 2    Data sources and database structure

### 2.1    Observation-based data sources

The original BDBP v1.1.0.6 (Bodeker et al., 2013) was built from measurements obtained from several satellite instruments and from ozone profile measurements made at sites in the global ozonesonde network (Hassler et al., 2008). The satellite instruments were selected first according to their vertical resolution (<3km), and then for their temporal and spatial coverage. Exceptions to these criteria were only accepted if the measurements provide information in a measurement sparse region or time period, e.g. the troposphere, the polar regions, or the early 1980s. BDBP v1.1.0.6 therefore included measurements from

the Stratospheric Aerosol and Gas Experiments I and II (SAGE I and II), the Halogen Occultation Experiment (HALOE), the Improved Limb Atmospheric Spectrometer (ILAS and ILAS II), and the Polar Ozone and Aerosol Measurement II and III (POAM II and III). Measurements from these satellite instruments are also included in BSVertOzone. While measurements from the Limb Infrared Monitor of the Stratosphere (LIMS) were included in BDBP v1.1.0.6, they were excluded from BSVertOzone since it was assessed that the vertical resolution of the LIMS measurements did not warrant its inclusion in the

new database. The temporal and vertical coverage of the instruments providing data for incorporation into the BSVertOzone database are illustrated in Fig. 1. A more detailed description of these instruments, their measurement techniques, their temporal and spatial coverage, and their measurement uncertainties, can be found in Hassler et al. (2014). Except for SAGE II, the data quality screening for other instruments was performed in the same way as described in Hassler et al. (2008). In addition, all SAGE I ozone measurements with uncertainties greater than 300% were excluded. The SAGE II screening criteria were

updated and are described in more detail next.

In late 2012, an improved and updated version of the SAGE II data set was released (version 7.0; Damadeo et al. (2013)). This replaced the version 6.2 data set that was used in BDBP v1.1.0.6. Due to the change in data version, the quality screening for SAGE II had to be adjusted. The screening of SAGE II ozone measurements was performed following the suggestions of Wang et al. (1996) together with the modifications outlined in the SAGE II version 7.0 release notes. In addition, all ozone

values below 23km between 1 July 1991 and 1 October 1993 were excluded as those measurements were affected by the Mt Pinatubo eruption (as it was done in Hassler et al. (2008)).

As a final quality check, ozone values from all data sources were used to calculate monthly mean zonal mean climatologies at each level and latitude bin. If individual values in these latitude bins exceed the respective mean by $3\sigma$, they were excluded.

BDBP v1.1.0.6 covered the period 1979 to 2007 since the main satellite data sources for that database, i.e. SAGE II and

HALOE, ended in 2005. After 2005, only ozonesonde profiles where included in BDBP v1.1.0.6. To extend BSVertOzone to 2016, it was necessary to add new satellite measurements and additional ozonesonde profiles to the database. To ensure sufficient overlap with SAGE II and HALOE measurements, preference was given to instruments that provide measurements starting in 2005 or earlier and extend to the end of 2016. Although having a somewhat broader vertical resolution than the





other data sources used in the BDBP v1.1.0.6 database, the large data quantity and high data quality of the Microwave Limb Sounder (MLS) makes it an attractive target data source for measurements for incorporation into the BSVertOzone database. The usefulness of MLS ozone data has already been shown through its use in several other combined ozone databases, e.g. GOZCARDS (Froidevaux et al., 2008) and SWOOSH (Davis et al., 2016). Given this pedigree, MLS measurements were
included in the creation of the BSVertOzone database.

The MLS instrument sits on NASA's Aura satellite which was launched in mid-July 2004 and remains operational to date (see Fig. 1). MLS measures microwave emission from the limb of the Earth's atmosphere to provide measurements of a multitude of atmospheric trace gases including ozone and temperature. MLS retrieved ozone profiles cover the region from the upper troposphere to the mesosphere, i.e. from 261hPa to about 0.02hPa. The vertical resolution of the ozone profiles ranges from
2.5km to 3km in the stratosphere, but increases to up to 8km in the mesosphere. MLS provides about 3500 ozone profiles every day, with a near global coverage from 82°S to 82°N (Waters et al., 2006). Version 4.2 MLS data were incorporated into the monthly mean zonal means in BSVertOzone. More information about the MLS instrument can be found in Waters et al. (2006), and more information about the MLS ozone validation in Froidevaux et al. (2008) and Hubert et al. (2016).

The screening of the MLS ozone measurements is based on the official MLS v4.2 data description document provided by JPL
(http://mls.jpl.nasa.gov/data/v4-2_data_quality_document.pdf), and is similar to the ozone screening described in Froidevaux et al. (2008). The uncertainties on each ozone measurement, at each pressure level, which is more detailed than the uncertainty information included in the MLS v4.2 data description document, were provided by the MLS team (Lucien Froidevaux, personal communication, April 2017).

One of the main foci of BSVertOzone is tracing all sources of uncertainty from the individual measurements through to the
final monthly mean zonal mean ozone values. The uncertainty estimates that are provided with the individual measurements obtained from each satellite instrument shown in Fig. 1, are used as a starting point for the uncertainty treatment throughout the creation process of the different BSVertOzone data sets (see following Sections). These uncertainties normally include, amongst others, calibration errors, spectroscopy uncertainties, and uncertainties introduced by the use of an a priori profile.

Ozonesondes remain the only source of ozone measurements throughout the troposphere (see Fig. 1) and were obtained
from four different data archives, viz. the World Ozone and Ultraviolet Radiation Data Centre (WOUDC), the Network for the Detection of Atmospheric Composition Change (NDACC), the Southern Hemisphere Additional Ozonesondes network (SHADOZ), and the National Oceanic and Atmospheric Administration (NOAA). While the goal is that each ozonesonde site provides profiles along with uncertainty estimates (Ozonesonde Data Quality Assessment, O3S-DQA) (Tarasick et al., 2016; Deshler et al., 2017; Witte et al., 2017), most data files obtained for use in the generation of BSVertOzone still did not include
uncertainty estimates. We therefore continue to use the uncertainty estimates as described in Hassler et al. (2008). To extend BSVertOzone through to the end of 2016, and to ensure that the most recently available ozonesonde data are used, an updated data set of ozonesonde data was obtained from the mentioned data archives.



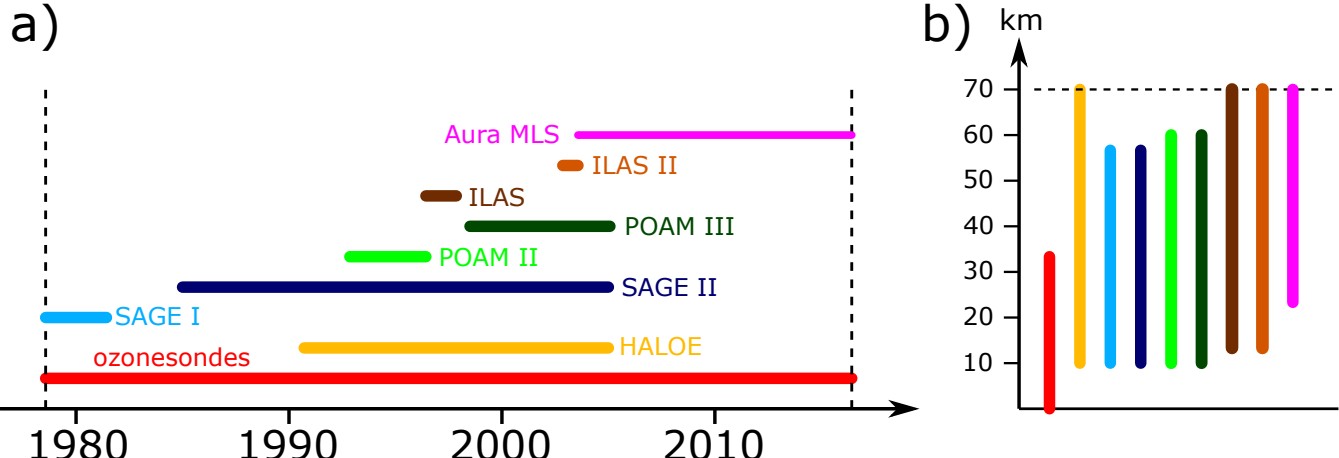

**Figure 1.** Temporal (a) and vertical coverage (b) of the different ozone data sources used to create the BSVertOzone database. Dashed lines in both parts of the figure denote the BSVertOzone database temporal (1979-2016) and vertical boundaries (0-70km).

## 2.2 Database grid

The monthly mean zonal means comprising the BSVertOzone database are provided in 5° latitude bands on a pressure grid and on a geopotential height grid. Ozone concentrations on these grids are provided both in number density and mixing ratio. To provide the input to those four databases, all individual measurements are converted from their original vertical grid and

5 concentration unit, to the vertical grids and concentration units prescribed in the BSVertOzone database. For these conversions, the temperature and pressure at each measurement is required. These ancillary data were available in the data files of the individual data sources. The BSVertOzone levels of the vertical altitude grid range from 1km to 70km with a spacing of 1km. The BSVertOzone levels of the vertical pressure grid range from about 878.4hPa to 0.046hPa and levels are approximately 1km apart (see Hassler et al. (2008) for more detail). Only a few satellite instruments provide data on such a grid and therefore,

in most cases, the measurements were interpolated onto the two pre-defined vertical grids. The uncertainties provided with each measurement are also linearly interpolated onto the pre-defined vertical grids using log pressure when interpolating between pressure levels. For simplicity, any reference to either a specific geopotential height or pressure level is hereafter referred to as "level".

## 3 Construction of the unfilled monthly mean zonal mean database (Tier 0)

Quantifying offsets and drift between different measurement systems can be made far more robust by using an independent data source, especially when temporal and spatial coincidences between the two measurement systems are sparse. If the independent data source has high spatial and temporal sampling and covers the combined range of the two measurement systems to be homogenized, it can be used as a transfer standard. The independent data source does not need to be quantitatively



exact but does need to capture the spatial and temporal morphology of the underlying measurements. Output from either a chemistry-climate model (CCM) that has been nudged towards observed meteorology, or output from a chemistry-transport model, can meet these requirements. The bias and drift correction applied here is based on a homogenization approach that uses a regression model together with global vertically resolved ozone concentrations as simulated by the chemistry-transport
model TOMCAT/SLIMCAT for the period 1980 to 2016 (see Fig. 2).

### 3.1 Chemistry-transport model (CTM) data

TOMCAT/SLIMCAT (hereafter referred to as SLIMCAT) is an off-line three-dimensional (3D) chemistry-transport model (CTM) (Chipperfield, 2006) that includes an interactive stratospheric chemistry scheme and is driven by meteorological fields obtained from ERA-Interim reanalyses (Dee et al., 2011) provided by the European Centre for Medium-Range Weather Fore-
casts (ECMWF). We use 12-hourly output from a SLIMCAT simulation at a horizontal resolution of $2.8°$ by $2.8°$, and 32 levels extending from the surface to about $60\,\mathrm{km}$ (corresponding to a vertical resolution of about 1.5-2 km in the stratosphere). Observed surface mixing ratios of source gases were used as boundary conditions for the SLIMCAT simulation. The known unphysical temporal discontinuities in ERA-Interim upper stratospheric temperatures in August 1998 that arose from changes in the satellite radiance data used in the assimilation (Dhomse et al., 2011), introduced a bias in the ozone concentrations
simulated by SLIMCAT in 1998. This bias only occurs at the top five model levels. Therefore, before the CTM data are used in this study, we remove the bias at those levels by first fitting a regression model that includes an offset term (whole time period) and step function (heavy side function) after 1998 to the ozone data (Dhomse et al., 2011). We then subtract the contribution of the step basis function from the model data, obtaining the corrected CTM data where the discontinuities have been removed. For more details about the SLIMCAT model see Chipperfield (2006), and Chipperfield et al. (2015, 2017).
Using CTM output as an evaluation and adjustment tool for coarsely distributed global ozone measurements is not a novel idea. In Sofieva et al. (2014) the gap-free ozone fields of a highly temporally and spatially resolved CTM run were used to characterize sampling biases for coarse satellite samplers when their measurements were used for the calculation of monthly mean zonal mean ozone values. Hegglin et al. (2014) used a CCM that was nudged to ERA-Interim reanalysis to correct for offsets and drifts between stratospheric water vapor measurements from multiple satellite instruments. This is very similar to
the approach used in our study. While Hegglin et al. (2014) used the CCM output to adjust monthly mean values of the different instruments, here the SLIMCAT output is used to adjust individual measurements with a correction that is based on zonal mean comparisons (see Sect. 3.2).

### 3.2 Homogenization

As shown in several recent studies (Hubert et al., 2016; Tegtmeier et al., 2013), satellite instruments can experience drifts over
their lifetime, and coincident measurements of the same constituent from different satellite-based and ground-based instruments can differ. It is necessary to account for such potential differences when combining measurements from different measurement platforms into a single product such as a monthly mean zonal mean. Two such approaches have been used recently within the community to combine measurements from different platforms. In both cases a standard is selected to which the measurements





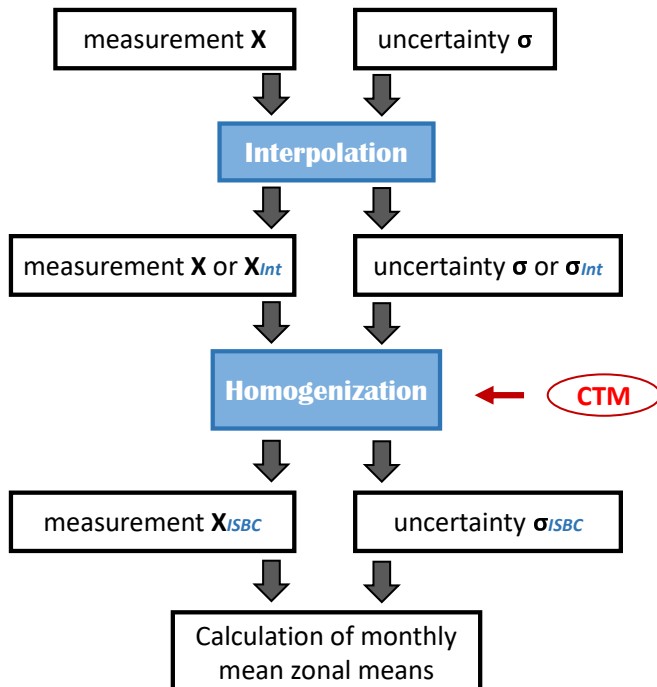

**Figure 2.** Flow chart describing the different modification and adjustment steps that are applied to the ozone measurements before they are used in the monthly mean zonal mean calculation. Note that 'ISBC' refers to inter-satellite bias correction, which is described in Sect. 3.2.

from other sources are adjusted. Preferably, this initial standard is sufficient in its global and temporal coverage to allow robust estimates of biases and drifts of all other measurement sets. Here we follow the quality assessment from Hubert et al. (2016) and the suggested ozone standard from Davis et al. (2016) and, for levels above 15km, chose measurements from SAGE II as the standard. Recognizing the higher data quality of the ozonesonde measurements below 15km, ozonesonde data are used at

5  15km and below as the standard. As each ozonesonde is individually prepared and calibrated, and most soundings are vertically integrated and validated against an independent total column ozone measurement, ozonesonde measurements are recognized as being more reliable than satellite-based measurements at lower levels. Once the standard has been selected, it is then necessary to either (i) find coincident measurements from the standard data set and from the measurement system to be adjusted, as done for example in Davis et al. (2016), or (ii) use an independent and gap-free data set that can be used as a transfer standard

10  for adjusting measurements to the standard. The first approach is problematic if a measurement from the chosen standard is not available for every measurement from the data set to be adjusted. For the second approach, reanalysis data, as shown, for example, by Foelsche et al. (2011), or a temporally and spatially highly-resolved output from of a CTM, as shown, for example, by Toohey et al. (2013) and Sofieva et al. (2014), can be used as a transfer standard, capturing small-scale ozone variability. Here we chose the second approach for generating a homogeneous data set, since measurements from some data sources are





limited to a very small geographical region and time period, such that coincidences with the SAGE II measurements, the chosen standard, are sparse. A CTM (SLIMCAT, see Sect. 3.1) simulation was used as the transfer standard.

The homogenization of the satellite-based measurements that contribute to BSVertOzone is a sequential process where a selected satellite instrument is adjusted with respect to the standard, hereafter referred to as the "inter-satellite bias correction",

ISBC (see Fig. 2). After the measurements have been adjusted to the standard, they are merged with the standard to produce a new standard with greater spatial and temporal coverage to which the measurements from a different satellite will be adjusted to and then merged. This process is repeated until all satellite-based or ozonesonde measurements are adjusted and merged with the standard. The order in which the satellites are chosen for ISBC is determined by the satellite that has the largest temporal overlap with the standard. In our case, above 15km where SAGE II is the standard, measurements from HALOE are

first adjusted and merged with the standard as HALOE provides the largest temporal overlap with SAGE II measurements. Below and at 15km, measurements from SAGE II are first adjusted and merged with ozonesondes. To adjust measurements from a different source to the standard, the following steps are taken at each level individually:

1. Calculate differences between individual ozone measurements from the standard and the CTM simulated ozone values.

2. Calculate an error weighted, latitude weighted (based on 1°latitude bands), monthly mean zonal mean of those differ-
ences.

3. Fit a linear regression model to the calculated monthly mean zonal mean differences (hereafter referred to as "modeled differences") to obtain an analytical representation of the difference field that can be evaluated at any latitude and time; $((standard - CTM)_{modeled}(\phi, z, t)$ at latitude $\phi$, level $z$ and time $t)$. The regression model comprises an offset and trend term, where the fit coefficients are expanded in Fourier and Legendre polynomials to account for seasonal and
latitudinal structure in the monthly mean zonal mean differences respectively. While the known discontinuities in ozone at the upper levels of the CTM data were corrected for to the extent possible, recognizing that some small inconsistencies may remain, it was decided to exclude the trend term from the regression model when modeling the difference fields for the levels above 47km (where the CTM data were corrected).

4. Repeat steps 1 to 3 using the measurements from the target new data source that requires bias correction to obtain an
analytical representation for its difference field; $(satellite - CTM)_{modeled}(\phi, z, t)$

5. Calculate an adjusted measurement at the location and time of a given satellite measurement using:

$$O_{3,adj}(\theta, \phi, z, t) = O_{3,raw}(\theta, \phi, z, t) + O_{3,ISBC}(\phi, z, t) \tag{1}$$

with

$$O_{3,ISBC}(\phi, z, t) = ((standard - CTM)_{modeled}(\phi, z, t)) - ((satellite - CTM)_{modeled}(\phi, z, t)) \tag{2}$$

where $\theta$ is the longitude, $O_{3,adj}(\theta, \phi, z, t)$ is the homogenized/adjusted measurement and $O_{3,ISBC}(\phi, z, t)$ is the applied adjustment.



6. Merge the adjusted measurements $O_{3,adj}(\theta, \phi, z, t)$ with the standard measurements, to create a new homogenized standard that now includes more measurements, and likely a larger spatial and temporal coverage, for the next iteration.

7. Repeat steps 1 to 6 for all data sources to be included in the BSVertOzone database.

The number of Fourier and Legendre polynomial expansions used to model the monthly mean zonal mean differences depend

on the individual differences provided as input to the regression model, as each satellite instrument provides measurements with a different spatial and temporal coverage. The number of Fourier and Legendre polynomial expansions used to model the difference field is made adaptive to avoid overfitting of the model. Four Fourier pairs and eight Legendre polynomials are the default expansions for the offset term while four Legendre polynomials and no Fourier pairs are the default values for the trend term. If the chosen default expansions result in overfitting of the difference fields, which is determined by testing whether the

maximum and minimum values of the modeled field do not exceed 150% of the maximum and minimum value of the original data, a new candidate model is generated by decreasing the degree of one of the expansion (e.g. one scenario would have three instead of four Fourier pairs for the offset term, with the rest of the offset and trend expansions remaining unchanged). The Akaike Information Criterion (AIC; deLeeuw, 1992; Bozdogan, 1987) was used to provide a relative assessment of the quality of the candidate models. The candidate model which minimizes AIC is selected for use, as it represents the model

which minimizes the amount of information lost. The process is repeated until a set of expansions is found that does not result in overfitting of the model.

In the generation of a homogenized data set, in addition to adjusting measurements from different measurement systems to account for bias and drifts, the uncertainties on those measurements also need to be revised since the application of these adjustments introduces additional uncertainty. Following error propagation rules, the uncertainty on the adjusted measurements

$\sigma_{adj}(\theta, \phi, z, t)$ is given by:

$$\sigma_{adj}(\theta, \phi, z, t) = \sqrt{\sigma_{raw}^2(\theta, \phi, z, t) + \sigma_{ISBC}^2(\phi, z, t)} \tag{3}$$

where the uncertainty ($\sigma_{ISBC}$) reflects the uncertainties on the regression modeled fields $(standard - CTM)_{modeled}$ and $(satellite - CTM)_{modeled}$, i.e.:

$$\sigma_{ISBC}(\phi, z, t) = \sqrt{\sigma_{(standard-CTM)_{modeled}}^2(\phi, z, t) + \sigma_{(satellite-CTM)_{modeled}}^2(\phi, z, t)} \tag{4}$$

While this re-evaluation of the measurement uncertainties does not include the effects of uncertainties in the CTM output, the effects of CTM uncertainties on the adjustment of the satellite data are minor, since the CTM data are only used as a transfer standard and, as can be seen from Eq. 4, cancel out if the CTM bias is consistent at both measurement locations.

A bootstrap method (Efron and Tibshirani, 1986) is used to estimate the uncertainties on the modeled differences, $\sigma_{(standard-CTM)_{modeled}}^2$ and $\sigma_{(satellite-CTM)_{modeled}}^2$. For example, to calculate $\sigma_{(standard-CTM)_{modeled}}^2$, the following steps are executed:

1. Fit the regression model (as described above) to the monthly mean zonal mean differences between standard measurements and CTM data.

2. Subtract the regression model fit from the data to obtain the residuals.





3. For each data point in the residual signal, add a randomly sampled value from a normal distribution with a mean of zero and a standard deviation of the measurement uncertainty for the selected point. This step captures that there is uncertainty on the residuals by virtue of the uncertainty on the measurements.

4. For each original measurement point, randomly select one residual value and add it to the data point. Do this for all measurement points to generate a new difference data set which, while having the same underlying structure as the original signal, now has different random noise. Then fit the regression model again resulting in a new modeled difference field.

5. Repeat steps 2-4 many times (e.g. 200) to generate 200 estimates of the modeled difference field. Calculate the standard deviation of those 200 modeled difference fields to obtain the estimated uncertainty on the modeled difference field $\sigma^2_{(standard-CTM)_{modeled}}$.

Such a bootstrap approach encapsulates two sources of uncertainty which are present in the modeled field: the uncertainty from the fact that the chosen model is imperfect, and the uncertainty in the measurements. The standard deviation of the difference fields, derived from the ensemble of fields, is the uncertainty on the modeled difference field.

### 3.3 Calculating the individual monthly mean zonal mean values

To create a homogeneous database, each measurement and its uncertainty, on a specific level, and in a specific latitude band, is adjusted using the ISBC method described above. The uncertainties on the corrections applied are included in the total uncertainty for each individual data point. For the final calculation of monthly mean zonal means values, additional data filtering was applied, similar to the filtering described in Bodeker et al. (2013), e.g. SAGE II data were not used below 10 km (below 242.8 hPa for the pressure grid), and SAGE I data were not used below 18 km (77.44 hPa).

Monthly mean zonal mean ozone values obtained from different data sources pre- and post-homogenization, for an example level and latitude band, are shown in Fig. 3. The different data sources complement one another and the homogenization process reduces the spread between the different data sources, so that they are more consistent with each other. The average adjustment for each measurement from the different data sources ranges from -14.4% for MLS to 37.7% for HALOE at 182 hPa between 25°N and 30°N (the chosen level and latitude band in Fig. 3), i.e. MLS measurements have a positive bias compared to the standard, while HALOE measurements have a negative bias.

The time series of ozone in Fig. 3 are shown for illustrative purposes only since the method to calculate the monthly mean zonal means does not depend on pre-calculated monthly mean zonal means for each individual data source, but rather calculates the monthly mean zonal mean values from all available individual measurements at once. All available measurements $x_i (i = 1..N)$ and their uncertainties $\sigma_i$ for each latitude band and level are then used to calculate the error-weighted monthly mean:

$$\overline{x} = \frac{\sum_{i=1}^{N}((1/\sigma^2_{i\_new}) \times x_i)}{\sum_{i=1}^{N}(1/\sigma^2_{i\_new})} \tag{5}$$





**Figure 3.** Monthly mean zonal mean ozone mixing ratios from different data sources (color coded as shown in the legends) at 182hPa between 25°N and 30°N. Upper panel: Unadjusted time series and lower panel: adjusted, bias corrected, time series. For more details see text.



where $\sigma_{i\_new}$ represents not only the measurement uncertainty but also the confidence we have that each measurement can be seen as an estimator of the monthly mean, and $\sigma_{i\_new}$ is given by:

$$\sigma_{i\_new}^2 = \sigma_i^2 + (x_i - x_{i\_exp})^2 \tag{6}$$

with the expectation value $x_{i\_exp}$ being the unweighted monthly mean zonal mean. The uncertainty on the monthly mean zonal

mean value $\overline{x}$ is then calculated using:

$$\sigma_{\overline{x}} = \sqrt{\frac{1}{\sum_{i=1}^{N}(1/\sigma_i^2)} \times \frac{1}{N-1}\sum_{i=1}^{N}\frac{\sigma_{i\_new}^2}{\sigma_i^2}} \tag{7}$$

with $N$ being the number of measurements available for the chosen latitude band and level. The uncertainty on the monthly mean zonal mean calculated using Eq. 7 is sensitive to the magnitude of the uncertainties on each measurement but also to the variance in the measurements, which is not the case by using equations for calculating the uncertainty on monthly

means as provided in the current literature. As the uncertainty on the monthly mean zonal mean takes into account how many measurements were available to calculate the mean value, the uncertainty will be larger for mean values where fewer measurements were available compared to mean values based on more measurements. However, having only one measurement per latitude band available is not sufficient to calculate a monthly mean and its uncertainty. As a result, the requirement here is that there are at least six measurements per latitude band and level available to calculate a monthly mean zonal mean. If fewer

measurements are available, no monthly mean zonal mean will be calculated.

The calculated monthly mean zonal mean ozone time series, combining all measurements from different data sources, are shown in Fig. 4 as solid orange lines (unadjusted) and blue lines (adjusted) with their corresponding uncertainties. Despite the data gaps, the annual cycle and the interannual variability within the monthly mean zonal mean time series at the level and latitude band are apparent as shown in Fig. 3 and Fig. 4. Without a homogenization process and the resulting measurement

adjustment towards the standard, the monthly mean zonal mean time series would represent mostly the mean of the data sources that are available at a high spatial and temporal measurement density in a given month and latitude band, introducing a bias into the monthly mean zonal mean time series. The uncertainties on the monthly mean zonal mean ozone values are significantly smaller once MLS measurements are added to the merged data product, as there are thousands of MLS measurements available per day.

The homogeneous database of monthly mean zonal means constitute "Tier 0" of BSVertOzone.

## 4   Creating a global, filled data set with the full range of variability (Tier 0.5 data set)

To generate a global, gap-free monthly mean zonal mean ozone data set, all values from the "Tier 0" database are used, and the missing monthly mean zonal mean values are estimated from correlations against a total column ozone database. This "Tier 0.5" data set includes the full range of measurement variability and is created as an intermediate step for the calculation of the

"Tier 1" data where a least squares regression model is used to attribute variability to various known forcing factors for ozone (Sect. 5).





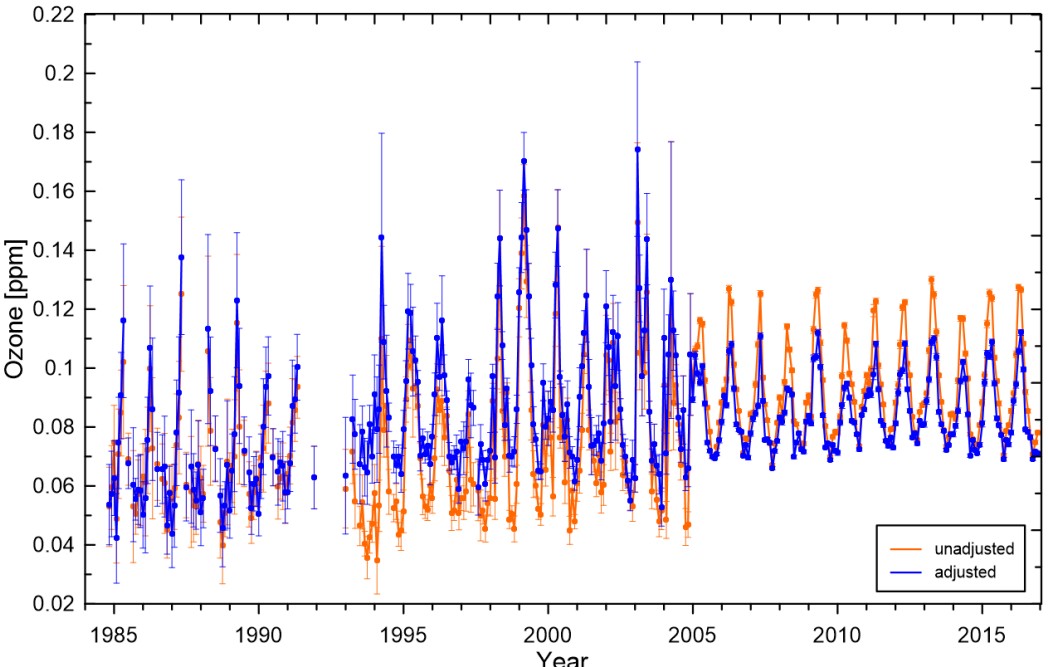

**Figure 4.** Monthly mean zonal mean ozone mixing ratio at 182hPa between 25°N and 30°N as calculated from all data sources before (orange line) and after (blue line) the homogenization process has been applied to the source data. For more details see text.

The first step in creating the Tier 0.5 data set is to regress the monthly mean zonal mean ozone at 20km/58.2hPa against monthly mean zonal mean total column ozone (TCO), i.e.:

$$O_3(m,\phi) = \alpha(m,\phi) \times TCO(m,\phi) + \beta(m,\phi) + R(m,\phi) \qquad (8)$$

where $m$ is the month, $\phi$ is the latitude, $\alpha$ and $\beta$ are the two regression model fit coefficients, each expanded in Fourier and

Legendre series, and $R$ are the residuals (see Bodeker et al. (2013) for details).

The TCO database used here is described in detail in Bodeker et al. (2018) and covers the period from 31 October 1978 to 31 December 2016 by combining measurements from multiple satellite-based instruments to a single global daily time series of ozone fields. Comparisons of TCO from a subset of satellite-based instruments against TCO measurements from ground-based Dobson and Brewer spectrophotometer networks are used to remove offsets and drifts between satellite-based and ground-

based measurements. For more details about the database and correction method see Bodeker et al. (2018). Daily total column ozone fields at 1.25°longitude by 1°latitude are then used to calculate monthly mean zonal means at 5°latitude bands before Eq. 8 is applied to pre-fill the vertically resolved ozone data set.

The regression model fit coefficients in Eq. 8 are expanded in Fourier series to account for seasonality in the correlation of ozone at 20km/58.2hPa against total column ozone, and in Legendre expansions to account for the latitudinal structure.

Uncertainties on the monthly mean zonal mean ozone values at 20km are passed to the regression model when the fitting is





performed. The optimal number of Fourier and Legendre expansions is determined by finding the minimum in a Bayesian Information Criterion (BIC; Liddle (2007)) for a range of possible Fourier (0..5) and Legendre (0..11) expansion indices. For example, at 20km this test identifies optimal values of four Fourier expansions and four Legendre expansions. The regression model can then be used to estimate missing ozone values, along with their uncertainties as derived from the uncertainties on

the regression model fit coefficients, whenever a measurement is not available. In this way, a filled monthly mean zonal mean ozone field at 20km/58.2hPa is created. This field then becomes the predictor (rather than TCO) for the field at 21km. Once filled fields at 20 and 21km are available, both are used as predictors for ozone at 22km, i.e. the regression model now includes two basis functions:

$$O_3(m,\phi,n) = \alpha \times O_3(m,\phi,n-1) + \beta \times O_3(m,\phi,n-2) + \gamma(m,\phi) + R \qquad (9)$$

Equation 9 is then applied from level 22 upwards to level 70, and then down from level 19 (now using levels 20 and 21 as the predictors for ozone at level 19) to level 1.

The result is a pre-filled ozone data set, where filled values include some indication of the true month-to-month variability as suggested by the TCO month-to-month variability. Monthly mean zonal mean ozone values at 20km after the homogenization of data sources has been applied (Fig. 5a) and the pre-filled data set are shown in Fig. 5.

This Tier 0.5 data set was then used as input to a least squares regression model to generate the Tier 1.1 to Tier 1.4 data sets described in the next section. It describes the full natural variability and is therefore particularly useful for CCM evaluation studies when the model runs with prescribed dynamics.

## 5 Creating global, filled data sets with only forced variability (Tier 1.x ozone data sets)

The methodology to generate the Tier 1.1 to Tier 1.4 data sets is much the same approach as the one described for previous

version of the database (BDBP v1.1.0.6) in Bodeker et al. (2013). The same regression model is applied, but rather than using this regression model to both fill the data gaps and conduct the attribution, it is now only used to conduct the attribution to various factors affecting ozone. This is a way to provide different data sets for very specific purposes with regards to CCM evaluation. For an in depth description of the approach see Sect. 4 of Bodeker et al. (2013). The most important features of the used approach are briefly outlined below.

The least square regression model that was applied to the Tier 0.5 data consists of eight basis functions, viz:

1. A constant offset that is expanded in a Fourier series to represent the mean annual cycle,

2. An EESC (equivalent effective stratospheric chlorine) term that differs with age of air,

3. A linear trend term,

4. A quasi-biennial oscillation (QBO) basis function that was specified as the monthly mean 50hPa Singapore zonal wind.

5. A second QBO basis function, that is mathematically orthogonalized to the first, to account for QBO lag variations with latitude and level,



**Figure 5.** Monthly mean zonal mean ozone mixing rations at 20hPa for a) unfilled adjusted ozone values from different data sources (Tier 0) and b) pre-filled (Tier 0.5) ozone database. For more details see text.



6. An El Nino–Southern Oscillation (ENSO) term,

7. A solar cycle term, and

8. A Mt. Pinatubo term that accounts for the enhancement of stratospheric aerosols after the Mt. Pinatubo eruption in 1991.

The regression model was applied to each level separately, where all basis functions are included for all levels, except the
EESC basis function which is excluded at levels below the mean tropopause. A two-term autocorrelation model was used to
account for the effects of autocorrelation within the monthly mean time series. The regression was applied to monthly mean
zonal mean values at all latitude bands simultaneously. This can be done by expanding the regression model fit coefficients
not only in a Fourier series to account for temporal variability, but also in Legendre polynomials to account for latitudinal
variability. The number of Fourier pairs and Legendre terms for each level differ, and is determined by the expected latitude
and time distribution of ozone for that level.

Similar to Bodeker et al. (2013), four different Tiers 1.x data sets were constructed as followed:

- Tier 1.1 (Anthropogenic): This data set is calculated by summing up the contributions from the offset, EESC and linear
  trend basis functions.

- Tier 1.2 (Natural): This data set is calculated by summing up the contributions from the offset, QBO, ENSO and solar
  cycle basis functions.

- Tier 1.3 (Natural and volcanoes): This data set is calculated by summing up the contributions from the offset, QBO,
  ENSO, solar cycle, and Mt. Pinatubo volcanic eruption basis functions.

- Tier 1.4 (All): This data set is calculated by summing up the contributions from all basis functions.

As the Tier data sets are output from a regression model, they do not capture real-world year-to-year variability, only the
variability for which basis functions are included in the regression model. These data sets are optimized for the use in com-
parisons with CCM simulations that do not exhibit the same unforced variability as reality. They can be used for different
purposes, e.g. to compare ozone radiative forcing with and without the effects of changes in EESC and greenhouse gases on
ozone.

Two examples of the Tier 0 database, and Tier 0.5 and Tier 1.x data sets for the latitude bands 30°S to 35°S, and 70°N to 75°N
are shown in Fig. 6 and Fig. 7. As there were no ozonesonde profiles available for the latitude band 30°S to 35°S (Fig. 6), there
are no measurements below about 250 hPa included in Tier 0. The stratosphere and lower mesosphere are covered well with
satellite measurements from around 1985, but the effect of the Mt Pinatubo eruption on the ozone measurements can clearly
be seen in around 1991 to 1993 in the lower stratosphere. More ozonesonde measurements and less satellite observations are
available for the latitude band 70°N to 75°N (Fig. 7), and the inclusion of MLS measurements from 2005 onward significantly
increases the coverage of the upper stratosphere and lower mesosphere.

The Tier 0 and Tier 0.5 data sets both show considerably more variability than the Tier 1.4 data set as the regression model is
not capable of tracking all of the variability in Tier 0 and Tier 0.5, rather only the variability that can be described by the basis





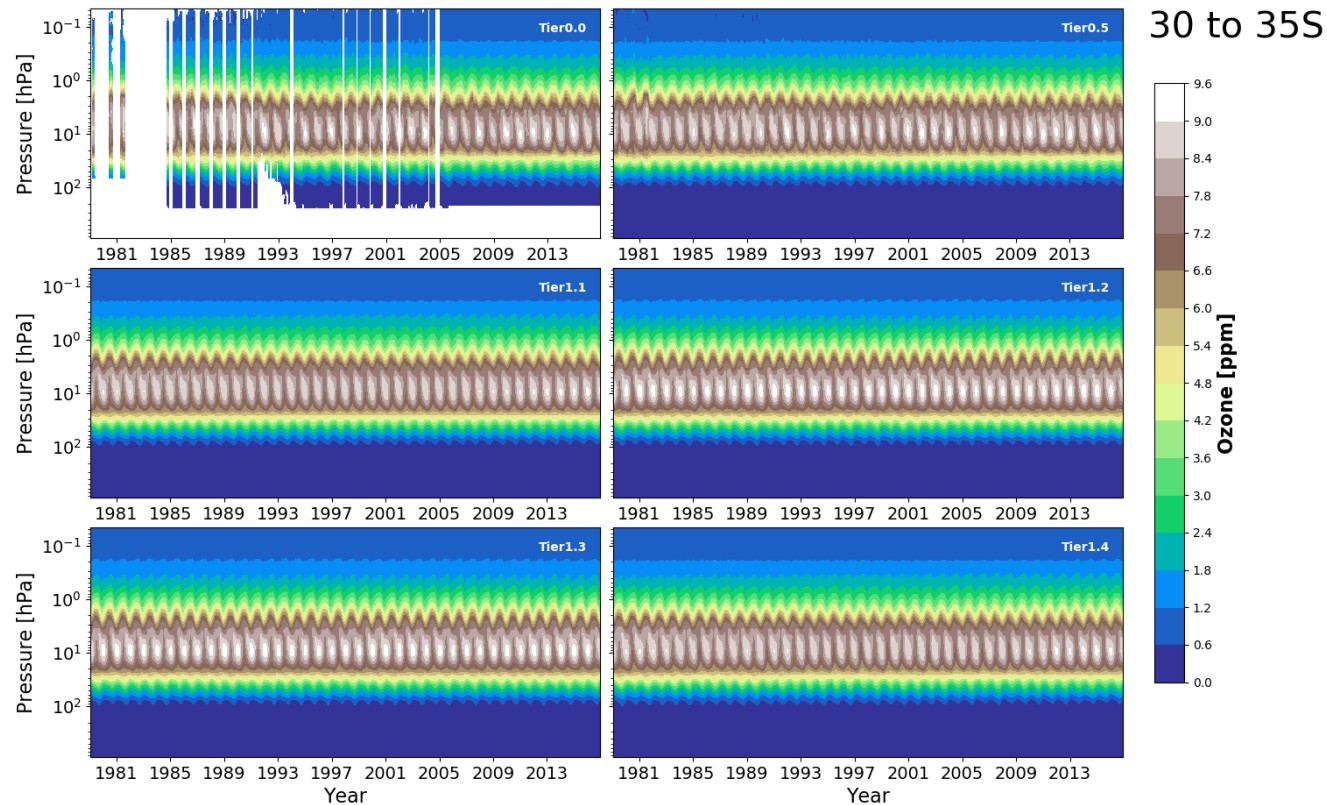

**Figure 6.** All six Tiers of BSVertOzone for the latitude band $30°$S to $35°$S (given in ozone mixing ratio on pressure levels).

functions used in the regression model. For example, in 2011 unprecedented depletion of ozone happened in the Arctic winter (Manney et al., 2011), which is detectable as less ozone compared to earlier years for pressure levels between 10hPa and 1hPa in Tier 0.5, but which is not apparent in Tier 1.4 (Fig. 7) since no basis functions are included in the regression model to track such variability. Careful comparison of the Tier 1.2 and Tier 1.4 data sets indicates the effects of natural variability on ozone, 5 while the comparison of Tier 1.1 with Tier 1.4 data set illustrates the effects of EESC on ozone.

## 6   Validation against other data sets

Due to the implemented improvements in the construction of the BSVertOzone database over the previous version BDBP v1.1.0.6, differences between both databases are to be expected. Ozone concentrations from 1979 to 2016 as extracted from Tier 1.4 of both databases at three different pressure levels and three different latitude bands are shown in Fig. 8. Ozone 10 concentrations in the upper stratosphere (10hPa) over the Northern mid-latitudes are almost identical (upper panel). Differences become apparent for ozone at around 140hPa in the tropics (middle panel): The ozone concentrations differ in magnitude, the





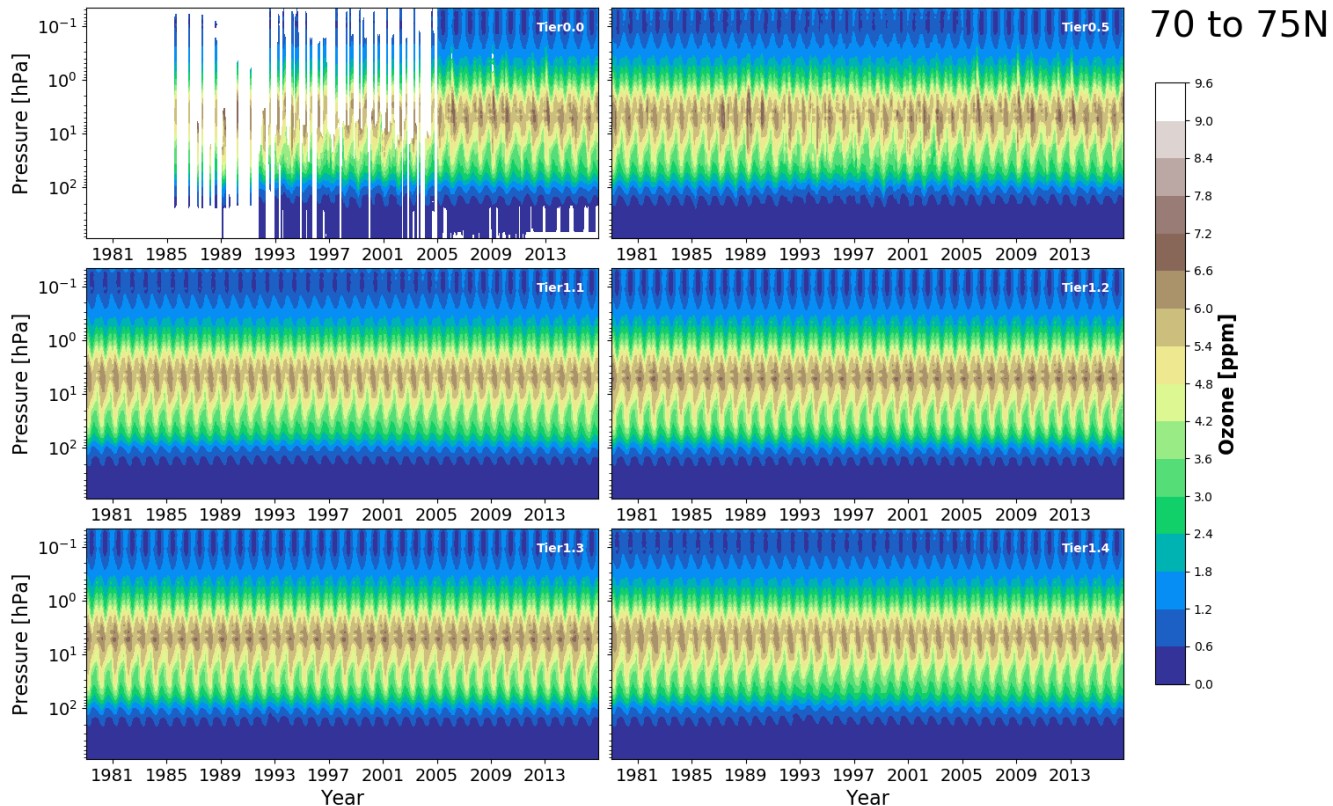

**Figure 7.** All six Tiers of BSVertOzone for the latitude band 70°N to 75°N (given in ozone mixing ratio on pressure levels).

BSVertOzone values (blue line) are consistently higher, especially after 1985, compared to the BDBP (green line). Furthermore, the annual cycle and interannual variability differ between BSVertOzone and BDBP, which could be a result of using different number of Fourier and Legendre expansions used in the regression model generating the Tier1.x data sets. While the expansions were pre-defined for the calculation of BDBP v1.1.0.6, the optimal number of expansions were determined by applying the BIC

5    method (see Sect. 3.2) when calculating BSVertOzone. At this pressure level, satellite-based measurements were adjusted to ozonesonde measurements (see Sect. 3.2), measuring higher ozone concentrations between 5°N and 10°N than satellite-based instruments. The BDBP includes the "raw" satellite-based and ozonesonde measurements, without any adjustment or correction being applied between the measurements from different instruments. Ozone concentrations at about 60hPa in the southernmost latitude band, 85°S to 90°S, as extracted from Tier 1.4 of BSVertOzone and BDBP are shown in the lower panel of Fig. 8.

10   Both data sets track the evolution of the Antarctic ozone hole in each Southern spring. However, there are small differences in the magnitude of the ozone reduction in the latitude band, especially at the beginning of the time series (1979 to 1985). The BSVertOzone (blue line) ozone concentrations are lower, e.g. there is more ozone depletion, than the ozone concentrations extracted from the BDBP v1.1.0.6 (green line) until about 1985. This is most likely a result of the homogenization approach



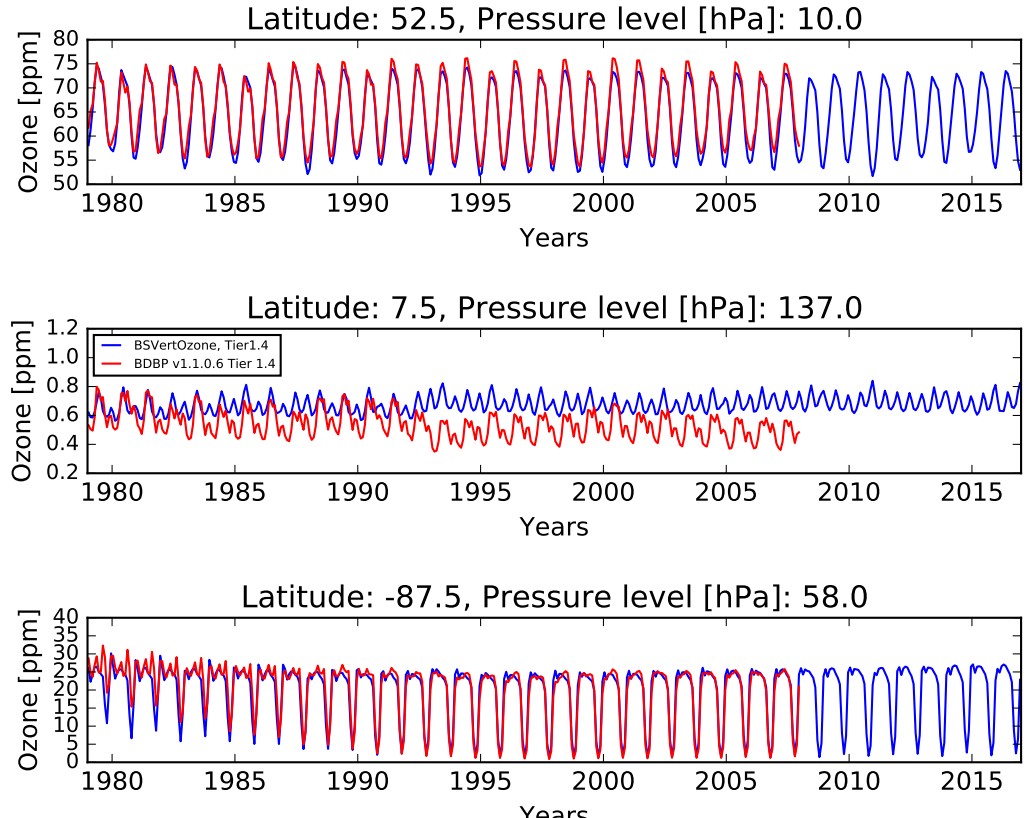

**Figure 8.** Ozone concentrations as extracted from BDBP v1.1.0.6 Tier1.4 (green line) and BSVertOzone Tier1.4 (blue line) for three different pressure levels and three different latitude bands.

applied to the different data sources in BSVertOzone but not in the BDBP. During the early 1980s, satellite-based measurements are sparse for most parts of the globe, and ozonesonde soundings were not as frequent as during the 1990s or 2000s.

Most measurements available for the latitude band and level described here (85°S to 90°S) are ozone soundings (South Pole station). There are only sparse satellite measurements available for this latitude band, and mostly later in the time period.
5 The homogenization adjustment for the South Pole ozone soundings therefore is only based on an extrapolated difference field between the standard and CTM (see 3.2). Problems related to this extrapolation could cause the slightly increased ozone depletion in the early part of the BSVertOzone Tier 1.4 time series compared to BDBP.

Davis et al. (2016) developed one of the more recent vertically resolved ozone data sets: SWOOSH (Stratospheric Water and OzOne Satellite Homogenized data set). This data set has already been used in a number of studies for a wide variety
10 of analyses (Harris et al., 2015; Steinbrecht et al., 2017; Ball et al., 2017). Therefore, SWOOSH is an ideal candidate for the evaluation of the newly developed BSVertOzone database. A comparison of ozone concentrations extracted from SWOOSH



**Figure 9.** Comparisons between SWOOSH (black line), BSVertOzone Tier 0.5 (blue line) and BSVertOzone Tier 1.4 (red line) for three different latitude bands and three different pressure levels.





with ozone concentrations extracted from Tier 0.5 and Tier 1.4 of BSVertOzone, for three selected different latitude bands and three different pressure levels, is shown in Fig. 9. As the SWOOSH ozone concentrations are provided on a different vertical grid, SWOOSH data were interpolated onto the BSVertOzone pressure levels for a direct comparison. Overall, the ozone concentrations from the Tiers of BSVertOzone (red and blue lines) agree well with the ozone values from SWOOSH

(black line) for the shown latitude bands and pressure levels. Some differences are apparent in the lower stratosphere of the Southern mid-latitudes (middle panel), where ozone concentrations from SWOOSH are higher than ozone concentrations from BSVertOzone. These differences are for the most part a result of the different homogenization approaches applied in SWOOSH and BSVertOzone. While in SWOOSH, all satellite-based measurements are adjusted to SAGE II measurements, in BSVertOzone all satellite-based measurements are adjusted to ozonesonde measurements at this pressure level (see Sect. 3.2).

The lower panel of Fig. 9 shows some small differences at around 70hPa over the tropics, where ozone concentrations from SWOOSH are occasionally higher than the ozone concentrations from BSVertOzone. These small differences are most likely a result of applying a different methodology when combining the measurements from different data source. Additionally, the selection of data sources could explain some of the differences seen in Fig. 9: SWOOSH does not include ozonesonde measurements, as well as SAGE I, POAM-II/III and ILAS-I/II measurements, whereas BSVertOzone does not include any

measurements from SAGE III and UARS MLS. In the middle to upper stratosphere, due to the good coverage of satellite data, selecting different of data sources will not affect the combined data product. However, in the lower stratosphere and upper troposphere where satellite measurements are sparse and more uncertain and ozonesonde measurements are more robust, not including ozonesonde measurements will lead to differences, as can be seen in the comparison between SWOOSH and BSVertOzone (Fig. 9 middle panel).

Comparisons between BSVertOzone Tier 0.5 and Tier 1.4 are in very close agreement, as would be expected. Tier 0.5 shows more interannual variability since its missing monthly mean zonal mean values are filled with regression model output describing the relationship between monthly mean ozone values from Tier 0 and monthly mean total column ozone values (see Sect. 4), and Tier 1.4 is smoother and shows less variation. Overall, however, time series of both Tiers track each other closely.

## 7   Discussion and conclusions

An updated and further developed version of the vertically resolved ozone database, the BDBP v1.1.0.6 (Bodeker et al., 2013), is presented in this study. Like its predecessor, the new database, BSVertOzone, consists of global monthly mean zonal mean ozone values between 1km and 70km (878.4hPa to 0.046hPa), and has several gap-free data sets. Monthly mean zonal mean ozone concentrations are provided in mixing ratio and number density for the period from 1979 to 2016 (Note: although only examples for the mixing ratio on pressure level data sets were shown throughout the paper, the other three databases

are also available through the doi given below). The BSVertOzone database is unique within the collection of available vertically resolved ozone datasets (e.g. GOZCARDS, SWOOSH and SAGE-CCI-OMPS) as BSVertOzone includes ozone profile measurements that cover the troposphere and the lower/mid stratosphere. Offsets and drifts between the measurements from different instruments are now quantified and accounted for by applying a homogenization method that uses a CTM output as a





transfer standard (see Sect. 3.2) and a regression model to adjust the measurements from different sources to a given standard. For levels above 15km, measurements from SAGE II are the chosen standard, while ozonesonde data are used at 15km and below as the standard (recognizing the higher data quality of the ozonesonde measurements below 15km). Ozone concentrations from BSVertOzone compare well to the ozone values from SWOOSH in most latitude bins, some discrepancies between the

two data sets remain. These can be explained by the differences in the methodology of combining measurements from different data sources. The applied homogenization results in an improvement of BSVertOzone compared to the earlier version BDBP v1.1.0.6, i.e. a more realistic representation of the ozone variability in the atmosphere.

As for the BDBP, BSVertOzone provides different Tier data sets (Bodeker et al., 2013):

- Tier 0 contains the monthly mean zonal mean values that are directly calculated from the individual (adjusted) data
sources; containing data gaps where no measurements were available.

- Tier 0.5 monthly mean zonal means represent an intermediate filled data set that is calculated from Tier 0 data. Missing monthly mean zonal mean ozone values are filled with regression model output describing the relationship between monthly mean ozone values from Tier 0 and monthly mean total column ozone values obtained from the total column ozone database described in Bodeker et al. (2018). The gap-free Tier 0.5 database captures real-world forced and un-
forced variability, suitable for CCM evaluations where the model was run with prescribed dynamics. Tier 0.5 is used as input for the calculation of the "Tier 1" data sets.

- Tier 1.1 to Tier 1.4 are based on multiple-linear regression model output. They differ in the combination of the contributions of the different basis functions used in the regression model. The ozone variability in these data sets is reduced compared to Tier 0 and Tier 0.5, since it describes only the variability for which basis functions were included in the
regression model. Especially Tier 1.4 is therefore well-suited for evaluating CCM output, where the CCM is not nudged to real-world dynamics.

A clear improvement compared to BDBP v1.1.0.6 is the provision of uncertainty estimates on each monthly mean zonal mean for all Tiers. These uncertainties combine the uncertainties that are provided with each individual measurements and the uncertainties introduced by applying the homogenization method. The provided uncertainties are essential for more realistic
comparisons with CCM simulations, and results of ozone variability analyses can be interpreted with more information about the confidence in the results.

There are several improvements that could be implemented when preparing the measurements and for the used homogenization method. In the current version (v1.0) of BSVertOzone, the global troposphere is only covered by ozonesonde profile measurements. These profiles are available for many decades (see Sect. 2.1), but they only cover a limited portion of the
globe. As a result, estimating the long-term global tropospheric ozone distribution from these measurements alone using the gap-filling method described above results in larger uncertainties in the troposphere than the stratosphere. Additional measurements in the troposphere would help to better constrain the regression model used to fill the data-gaps and therefore reducing the overall uncertainties on the monthly mean zonal mean ozone values. Since the early 2000s there are satellite measurements





of vertically resolved tropospheric ozone available (e.g. the Tropospheric Emission Spectrometer (TES), or the Atmospheric Infrared Sounder (AIRS)) which could be included in the database. Although both satellite instruments only provide measurements for the last 15 years, the additional information would improve the characterization of the global tropospheric ozone concentrations of the database covering the full period.

Measurements from MLS are the only source of stratospheric ozone data in the last 10 years which were included in the current version of the database. As long as MLS is active and measures ozone, BSVertOzone will be updated regularly to include these MLS measurements. However, when MLS stops measuring ozone, alternative, and possibly additional, data sources for stratospheric ozone will need to be added to BSVertOzone to ensure a continuous time series of vertically resolved ozone into the future. Measurements from NASA's Ozone Mapping Profiler Suite (OMPS), from SCISAT's Atmospheric

Chemistry Experiment-Fourier Transform Spectrometer (ACE-FTS), or from the recently launched SAGE III instrument on the International Space Station (ISS) would be possible candidates to be included in BSVertOzone.

Besides including more ozone measurements from different instruments, there are some planned improvements in the processing of the measurements that are planned to be implemented in the future. Firstly, as the CTM output used here as a transfer standard to homogenize the satellite and ozonesonde measurements, has a temperature bias due to the underlying meteorological

ical ERA-Interim reanalysis (see Sect. 3.1), it is planned to use CTM output forced with the most recent reanalysis data set ERA5. This most likely will remove the remaining inconsistencies in the ozone concentrations simulated by SLIMCAT in the upper levels.

All available measurements for each latitude band, each level and each month are most likely not evenly distributed spatially and temporally, which can result in a skewed (non-representative) monthly mean value, and an underestimation of the monthly

mean uncertainty. The individual ozone measurements should therefore undergo a spatial and temporal bias correction before monthly mean zonal means are calculated, to represent the monthly distribution correctly. Additionally, it might be necessary to consider possible existing spatial and temporal autocorrelations between individual data points. As mentioned in Section 5, a two-term autocorrelation model was used in the regression model generating the Tier 1 data sets. This only takes into account the temporal dependence of the already calculated monthly mean zonal mean values, but does not correct for the temporal

dependence of measurements within one month, or any spatial dependence of measurements within the chosen latitude band. While this consideration would not affect the calculated monthly mean zonal means, it would change the number of available independent measurements, and therefore would change the uncertainties on the calculated means.

In the upper stratosphere and mesosphere, ozone formation and destruction happens so fast that it follows the availability of sunlight. As a results, diurnal variations in ozone concentrations are observable in the upper stratosphere and lower mesosphere

(Schanz et al., 2014). Therefore, differences in ozone concentrations measured in the upper stratosphere/lower mesosphere could result from satellite-based instruments measuring ozone at a different solar zenith angles, i.e. different local time of the day (e.g. Pallister and Tuck, 1983; Schanz et al., 2014; Studer et al., 2014). With the current version of BSVertOzone, the potential differences in ozone measurements caused by the diurnal cycle are ignored as the effect on the monthly mean zonal mean ozone values it is expected to be small. However, to test this expectation, and for an update of BSVertOzone, it is planned

to implement methods to account for the diurnal cycle effects on the upper levels of the ozone database.

## 8    Data availability

BSVertOzone v1.0 that is described in this paper is archived and publicly available at Zenodo (Zenodo is a research data repository that was created by OpenAIRE and the European Organization for Nuclear Research (CERN)) with the DOI number: http://doi.org/10.5281/zenodo.1217184. Additionally, it is available at the Bodeker Scientific website (http://www.bodekerscientific.com/da

5    in its most recent and updated version.

*Author contributions.*  B.H. wrote the paper with support and input from all other co-authors. B.H., S.K., G.E.B. and J.L. were involved in combining the satellite measurements and performed the several computations required for the generation of the BSVertOzone database. K.N. was involved in developing the software for the homogenization method described in the paper. Comparison between the BSVertOzone and SWOOSH data sets was performed by S.M.D. The SLIMCAT simulations were performed by M.P.C and S.S.D, who also provided the

10    model output data for this study. M.D. assisted in writing the paper and provided valuable discussions around the methodology.

*Competing interests.*  The authors declare that they have no conflict of interest.

*Acknowledgements.*  This work was conducted under subcontract to NIWA under the Deep South National Science Challenge (CO1X1445). The SLIMCAT modeling work was supported by the NERC National Centre for Atmospheric Science (NCAS). We thank Wuhu Feng for help with SLIMCAT. The simulations were performed on the national Archer and Leeds ARC HPC facilities.



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
