# Peer review of "An updated version of a gap-free monthly mean zonal mean ozone database"

_Earth System Science Data, 2018_

## Referee Comment (RC1) · Anonymous Referee #1 · 3 Jun 2018

This paper describes a new version of the Bodeker Scientific ozone data record. The new version has been extended to 2016 and uses improved, or new methods to: account for bias and drift between data sources; choose the optimum number of Fourier and Legendre coefficients; trace measurement uncertainties through to the final database products.

The new database has been compared with SWOOSH and with the old Bodeker Scientific database and it compares well with SWOOSH and shows improvements over the previous data record. This paper is quite well written and can be published with only minor revisions.

Comments/questions:

[Figure]

Why weren't any of the European or Canadian satellite based ozone data records included within the database?

Page 6 – line 10: The choice to linear interpolate the uncertainties should be justified. Although is intuitively correct it can be argued that it is not statistically correct.

Page 9 – line 26: The definition of the term O3_raw in eq (1) is not obvious.

Page 10 – line 21: The definition of sigma_raw is also not obvious.

Page 15 – line 9: In eq (9) I assume that n is the level number. Can you better explain how this step is used to merge information from or transfer information between three altitude/pressure levels?

Figures 6 and 7 are not overly informative. The last four panels of these are visually identical on my printed copy. Perhaps a presentation of the differences between the Tiers might be more informative.

Typos

Page 4 – line 30: "where" should be "were". Page 15 – line 13: The text says 20 km while the figure caption says 20 hPa Page 17-line 19: the text should say "Tier 1.*x*". Page 22 – line 16: "of" should be removed.
* * *

---

## Referee Comment (RC2) · Anonymous Referee #2 · 7 Jun 2018

This manuscript describes the next generation of the Bodeker Scientific data set BDBP v1.1.0.6, known as BSVertOzone v1.0. Most notably this version includes AURA MLS data, a means of accounting for offsets and drifts between source data sets, and an explicit treatment of uncertainties. In addition the filling of gaps in the data set is now done independently of the full regression fit. The manuscript is well written and well resourced. The data set is a unique product, and access information for the final data product and all inputs are given in the manuscript. I have mostly minor comments/questions, and recommend publication after these issues have been addressed or clarified.

Minor Comments: Are both MLS ascending and descending profiles included? To that end, is there any attempt to account for diurnal ozone variations when combining the

target data sources? Does the CTM sufficiently capture day/night variations? Update: I see mention of this at the end of the article. However, does the CTM 12-hour resolution provide any diurnal information?

On the vertical coordinate transformations, do the various data sources provide the same pressure/temperature information? Would it be more consistent to use the same temperature/pressure data to do all the conversions rather than use the sources provided with each satellite data set, which may vary? Has the sensitivity to pressure/temperature fields been tested or considered as part of the measurement uncertainty (I may have missed this in previous papers)?

P9 On the description of the homogenization technique, I did get a bit confused. After reading it a couple of times, I think one problem is the word merging. I think of merging as going from multiple measurements to one in some fashion, but if I understand correctly you are adjusting individual measurements and then accumulating more and more measurements into the standard (as opposed to averaging monthly zonal mean fields at this stage).

If this is correct, I suggest the following wording tweaks be considered in lines 3-8 " . . . is a sequential process where each measurement from a selected satellite instrument is adjusted with respect to the standard, hereafter referred to . . . After the measurements have been adjusted to the standard, they are incorporated as part of the standard, and the new set of standard measurements is used to determine the adjustment for the next set of target measurements. This process is repeated until all satellite-based and ozonesonde measurements have been hmoginized.

P9 Step 2: I suggest using a bit more precise wording here. I first read it as area weighting measurements within a 1 degree band. What about "Calculate an error weighted monthly mean zonal mean of the differences at 1 degree zonal resolution, then scale each bin average by the cosine of the bin's central latitude. Is that what the authors meant? What is the error weighting, $1/sigma$ or $1/sigma^2$?

P10 Step 6: suggest Incorporate the adjusted measurements. . .

P11 In the bootstrap steps 3 and 4, I'm wondering if the order matters. It seems measurement uncertainty should be locked in time. That is, first the residuals should be randomly rearranged in time to represent a different noise structure (Step 4), then the measurement uncertainty should be added (Step 3). If a given instrument has a period of time where the measurement uncertainties are higher, those higher measurement uncertainties should occur at the same time, the time of the known issue. If the larger uncertainties occur over say a month, but then are randomly rearranged to occur at 30 random days over the domain, the integrity of the measurement uncertainty is lost.

I also got a bit confused as to what was being bootstrapped. The fit, and thus the residual, is to the monthly mean zonal mean difference field. Is the measurement uncertainty in bootstrap Step 3 that of an individual profile, or of a particular monthly mean zonal mean bin? If the latter (which I was thinking it should be), how is this uncertainty computed? At step 4 it seems to go back to the individual profile, but still the residual uncertainties being added are for monthly mean zonal mean values. If it truly is going back to the original data, the new data with the added uncertainties are then re-averaged into monthly mean zonal means, thus beating down some of that variability. Or is it just that each monthly mean zonal mean value is given a random residual to construct the new difference field, and then this new field is fit with the regression model? That approach makes more sense to me, to work with the monthly mean zonal mean values, rather than revert all the way back to individual profiles. It seems the month to month noise from the residual is being used to replicate the day to day noise within the month of the individual measurements.

P11 L20: In reference to Figure 3, it would be helpful to state what the original standard was at this level. You can see from the plot that sonde data didn't change, but adding it to the text would be good. Also, why are the sonde data so limited before 1995? How were the SAGE adjusted in this case, is the CTM used over a multi-year period to make matches between SAGE and sonde? Finally, in reference to Figure 3, is there

published literature to compare the stated offsets for MLS and HALOE, or do direct comparisons between HALOE and sonde/MLS and sonde back up these values? Even with diminished match-ups, a difference approaching 40% should be detectable.

P15, Eqn 9: Did the authors do any sensitivity testing with the regression fits to two levels, in particular in cases where the ozone at the two levels is highly correlated. I would think that often using fits to both levels is not needed, are there specific situations were the fit to two successive layers is particularly useful?

P18: The details in the differences in Tier 1.x are difficult to see. Maybe a particular year or event period can be highlighted, showing the full time series for Tier 0 and Tier 0.5, but a shorter sample period for the other data sets to point out specific features.

Did the authors look at the consistency between the integrated Tier 0 and Tier 0.5 and the total ozone? I think such a comparison would be useful as a validation point and to show the influence of using the total ozone to fill the data set. With a full profile data set and independent total ozone merged data set from the same group, users might naturally work with both simultaneously, so analyzing their consistency would be useful. This is a validation that the other merged data sets cannot easily do because they do not have full vertical resolution.

Technical Corrections: P2 L20 monhtly → monthly P3 L4 beside → besides (or in addition too) P3 L7 suggest wording change, maybe still have limited coverage in the troposphere P3 L11-12 suggest wording change, maybe "This is particularly important when seeking to detect the small but expected signal of ozone recovery due to reductions in ozone depleting substances." P3 L24 (Sect 2) P4 L30 where → were P5 L2 remove "for measurements" P7 L14 remove "," after Dhomse references P7 L17 heavy side function → Heaviside Function P8 L3 It would be helpful to include the approximate pressure corresponding to 15km here. P8 L6 I'm not sure of the meaning of the sentence starting "As each ozonesonde is individually prepared..." I would argue that the inherent physics of the satellite measurements vs sonde is the primary reason the

ozonesonde measurements are more reliable in the troposphere (not better calibration/validation). Or are you saying that we know sonde data are more reliable because of the very precise calibration/validation? In any case I think you could easily remove this sentence. It seems to repeat the sentence before, which stands on its own. P8 L12 '. . . ,or temporally and spatially highly-resolved output from a CTM, ...” P9L21 were corrected to the extent possible (remove “for”) P11 L2: suggest “This step represents the influence of measurement uncertainty on the residual.” P15 L24 remove “used” P19 L1 and Figure 8 Caption: green -> red P20 L3 (85deg S to 90 deg S, 58 hPa) P22 L1 remove “different” or “selected” P22 L12 different data sources P22 L16 remove “of” P23 L4 though some discrepancies P23 L23 each individual measurement P23 L29 and therefore reduce P24 L12 remove “planned” (used later in sentence) P24 L14 remove “,” after measurements

---

## Author Comment (AC1) · 27 Jul 2018

**Response to reviewer comments on "An updated version of a gap-free monthly mean zonal mean ozone database" by B. Hassler et al.**

We appreciate the suggestions and constructive comments provided by both reviewers. Below, the reviewer's comments are repeated in blue with our response in black.

**Response to Reviewer #1**

Comments/questions: Why weren't any of the European or Canadian satellite based ozone data records included within the database?

We carefully assessed measurements from all satellite-based instruments for inclusion in our database. In some cases, while we have planned to include specific data sets in our database, we have been unable to date as a result of resource constraints. The bulk of the work to construct this database took place at Bodeker Scientific, an organisation that relies exclusively on soft funding to conduct its research. This places constraints on time that can be spent on tasks such as this, with the result that it was not possible to be comprehensive in the data sets that we included in our database. We selected those data sets that provided the most value (long time series, large number of measurements per day, high vertical resolution, good latitude coverage, small measurement uncertainties). As more funding becomes available, more data sets will be included in the database.

Our assessment of data sets to include/exclude follows:
- ***Included:*** Aura MLS, ILAS, ILAS II, POAM II, POAM III, SAGE I, SAGE II, HALOE.
- ***Planned but not yet done:*** SAGE III, LIMS, ACE-MAESTRO, GOMOS, Odin-OSIRIS, OMPS, TES, AIRS.
- ***Record deemed too short to warrant inclusion:*** HIRDLS.
- ***Excluded because vertical resolution was too low:*** ACE-FTS, UARS-MLS, MIPAS, Odin-SMR, SMILES, SCIAMACHY, SBUV, GOME/GOME-2, OMI, IASI (note that in the paper we state that we selected for instruments that measure at 3\,km vertical resolution or better).

So we do still have several data sets that are planned to be added in future versions of the database when resourcing becomes available.

Page 6 – line 10: The choice to linear interpolate the uncertainties should be justified. Although is intuitively correct it can be argued that it is not statistically correct.

A lot of time and effort was spent investigating how best to interpolate measurement uncertainties to the common data grid. Linearly interpolating measurements is clearly suboptimal since one would expect the uncertainty half way between the two measurements to be larger than either of the single neighbouring uncertainties since we have less knowledge about the species distribution at that altitude. Any formula such as a weighted uncertainty estimate e.g.

$$\sqrt{a \times \sigma_1{}^2 + (a - 1) \times \sigma_2{}^2}$$

where $a = 1$ at measurement location 2 and $a = 0$ at measurement location 1, can result in the uncertainty between the two measurements being smaller than the uncertainty at either of the measurement locations. The statistically correct way to estimate the interpolation uncertainty is by means of Kriging and the use of a 1-dimensional variogram. See, for example, an explanatory figure here:
https://en.wikipedia.org/wiki/Kriging
This approach, however, is computationally very expensive for large data sets of individual ozone measurements whose variagram changes in space and time. Implementing a Kriging approach was beyond the compute power we have access to. We therefore simply linearly interpolated the uncertainties, acknowledging that this is an imperfect approach. To address the concern of the reviewer we have made the following change to the text:

+++++
The uncertainties provided with each measurement are also linearly interpolated onto the pre-defined vertical grids using log pressure when interpolating between pressure levels. We acknowledge this approach to interpolating the uncertainties under-estimates the uncertainties on the interpolated values since we expect uncertainties to maximize between measurements. A rigorous statistical approach to interpolating the uncertainties is by means of Kriging and the use of a 1-dimensional variogram. However, this approach is computationally demanding and, given that the variogram changes in time and space, and that the interpolation is generally over less than 1\,km, we assessed that the small under-estimation in the uncertainties on the interpolated values has little effect on the uncertainties in the calculated monthly mean zonal means which are largely dominated by spatial and temporal variables in the measurements.
+++++

Page 9 – line 26: The definition of the term O3_raw in eq (1) is not obvious.

We have included the definition of O3_raw (which is the original unadjusted measurement) in the text.

Page 10 – line 21: The definition of sigma_raw is also not obvious.

The term sigma_raw refers to the uncertainty on the original, unadjusted measurement and we have now clarified that in the revised manuscript.

Page 15 – line 9: In eq (9) I assume that n is the level number.

Correct, *n* refers to the level number. We clarified that in the revised manuscript.

Can you better explain how this step is used to merge information from or transfer information between three altitude/pressure levels?

Consider the case at level *n* where we now have filled monthly mean zonal mean fields at levels *n-1* and *n-2*. Where values are missing at level *n*, we use the values at levels *n-1* and *n-2* as predictors for the value at level *n*. See figure below. Note, however, that how the ozone value at level *n* depends on the ozone values at levels *n-1* and *n-2* (i.e. the determination of the three fit coefficients in equation (9)), is not determined individually for each month and latitude. Rather the fit coefficients are derived from a global fit of equation (9) across all months and years where the seasonality in the fit coefficients is captured by expanding them in Fourier series, and their latitudinal structure is capture by further expanding the fit coefficients in Legendre expansions. This prevents localised noise or anomalies in the data at levels *n-2* and *n-1* from affecting the estimates of the ozone values at level *n*.

[Figure]

We have not added any additional material to the manuscript since we believe that the explanation provided in the paper, while concise, provides the reader with all information required should they wish to replicate this method.

Figures 6 and 7 are not overly informative. The last four panels of these are visually identical on my printed copy. Perhaps a presentation of the differences between the Tiers might be more informative.

We agree with the reviewer that the differences between the different panels are difficult to see. Therefore we followed the reviewer's suggestion and included new Figure 6 and 7. They are now showing the differences between Tier1.x and Tier 0.5 data. We updated the figure caption accordingly.

Corrected.

Page 15 – line 13: The text says 20 km while the figure caption says 20 hPa

Corrected. 20 km is correct.

Done.

Done.

**Response to Reviewer #2**

Minor Comments: Are both MLS ascending and descending profiles included?

Yes, both ascending and descending profiles are included.

To that end, is there any attempt to account for diurnal ozone variations when combining the target data sources? Does the CTM sufficiently capture day/night variations?

We have been developing a methodology to account for diurnal variations in the target data sources, primarily at levels of 50 km/0.8 hPa and above. The method uses fits of the form:

$$O_3 = (c_0 + c_1 \times T) * erf((\theta - (a_0 + a_1 \times T)))/(b_0 + b_1 \times T) + d_0 + d_1 \times T + e_0 \times \theta_0 + e_1 \times \theta_0 \times T + f_0 \times \theta_1 + f_1 \times \theta_1 \times T$$

to 6-hourly CTM ozone concentrations plotted as a function of solar zenith angle (SZA) – see, for example, the plot below (fit are performed separately for am and pm). $\theta$ is the SZA and $a$ to $f$ are fit coefficients that depend on season (expanded in Fourier series)

and are derived individually for each 5° latitude zone. *T* is the temperature - including

the temperature effect improved the attribution of the ozone diurnal cycle to SZA. *erf* is the Error function see, for example https://en.wikipedia.org/wiki/Error_function Once the *a* to *f* fit coefficients for every latitude zone have been obtained from fits to 6-hourly SLIMCAT data, they can be used to analytically normalize the ozone concentrations to a common SZA - typically 60°. This SZA correction was not applied to this version of the database since the

quality of the fit of the equation detailed above, especially around the day-night terminator, was imperfect, i.e. there was structure in the ozone vs. SZA relationship that was not captured well enough by this fit. More work needs to be done to derive an SZA normalization for our database and this is slated for inclusion in the next version of the database.

Update: I see mention of this at the end of the article. However, does the CTM 12-hour resolution provide any diurnal information?

The SLIMCAT data are provided as 12-hourly snapshots in UTC such that at any latitude, the SZA varies with longitude and by sampling ozone across all longitudes, a wide range of SZAs is sampled. That said, just to be safe, we obtained 6-hourly SLIMCAT output for a single year to derive the necessary parameterisation of ozone as a function of SZA for am/pm for all latitudes and for all seasons.

On the vertical coordinate transformations, do the various data sources provide the same pressure/temperature information?

No they don't. For example, some satellite will have altitude as its native vertical coordinate and will provide retrieved temperature as an ancillary variable. Another satellite may also have altitude as its native vertical coordinate but will provide reanalysis temperature as an ancillary variable. Other satellites may have pressure as their native vertical coordinate and provide either retrieved or reanalysis temperatures as an ancillary variable. Those that provide ancillary variables from reanalyses may not use the same reanalyses. Where geopotential height is not provided in the data files, we calculate geopotential height (one of our two primary vertical coordinates) from pressure

and temperature profiles provided by the satellite retrieval teams or from geometric altitude if that is the native vertical coordinate for the instrument.

Would it be more consistent to use the same temperature/pressure data to do all the conversions rather than use the sources provided with each satellite data set, which may vary?

Possibly, but where the instrument does retrieve e.g. its own temperature profile along with the ozone profile, we thought it better to use that temperature profile to convert from e.g. the native altitude coordinate for the retrieval to a pressure coordinate as the satellite temperature profile is expected to be more accurate that data extracted from reanalyses. In cases where the instrument does not retrieve its own temperature profile, we used the profile provided by the instrument team. Yes we could have used data from the same reanalyses in all such cases. This is a good suggestion from the reviewer and we will test that idea when we create the next version of the database.

Has the sensitivity to pressure/temperature fields been tested or considered as part of the measurement uncertainty (I may have missed this in previous papers)?

No it wasn't. To some extent we would expect that the instrument retrieval teams would include that contribution in their estimate of the total uncertainty on each measurement. We acknowledge that this is not the case but rather than us trying to figure out if and when this contribution has been considered in the measurement budget, we are working with the satellite retrieval teams to encourage them to include *all* sources of uncertainty in the uncertainty budgets so that the uncertainties provided in the data files are true uncertainties.

P9 On the description of the homogenization technique, I did get a bit confused.

That's OK. So did we - regularly.

After reading it a couple of times, I think one problem is the word merging. I think of merging as going from multiple measurements to one in some fashion but if I understand correctly you are adjusting individual measurements and then accumulating more and more measurements into the standard (as opposed to averaging monthly zonal mean fields at this stage).

Yes, that is correct.

If this is correct, I suggest the following wording tweaks be considered in lines 3-8 "... is a sequential process where each measurement from a selected satellite instrument is adjusted with respect to the standard, hereafter referred to … After the measurements have been adjusted to the standard, they are incorporated as part of the standard, and the new set of standard measurements is used to determine the adjustment for the next set of target measurements. This process is repeated until all satellite-based and ozonesonde measurements have been homogenized.

We agree with this change of wording and have made the suggested changes.

P9 Step 2: I suggest using a bit more precise wording here. I first read it as area weighting measurements within a 1 degree band. What about "Calculate an error weighted monthly mean zonal mean of the differences at 1 degree zonal resolution, then scale each bin average by the cosine of the bin's central latitude. Is that what the authors meant? What is the error weighting, 1/sigma or 1/sigma^2?

The term used to calculate the weight of each measurement in the weighted monthly mean zonal mean is cos(latitude)/sigma^2.
We added the following text to the revised manuscript to clarify the meaning of Step 2:
"Calculate an error and area weighted monthly mean zonal mean of those differences at 1 degree zonal resolution. The weight of each measurement is defined as cos(latitude)/sigma^2.

P10 Step 6: suggest Incorporate the adjusted measurements…

We agree with the reviewer and changed the wording as suggested.

P11 In the bootstrap steps 3 and 4, I'm wondering if the order matters. It seems measurement uncertainty should be locked in time. That is, first the residuals should be randomly rearranged in time to represent a different noise structure (Step 4), then the measurement uncertainty should be added (Step 3). If a given instrument has a period of time where the measurement uncertainties are higher, those higher measurement uncertainties should occur at the same time, the time of the known issue. If the larger uncertainties occur over say a month, but then are randomly rearranged to occur at 30 random days over the domain, the integrity of the measurement uncertainty is lost.

First we should clarify that the bootstrapping is performed to obtain an estimate of the robustness of our regression model not to obtain the uncertainty on the monthly mean zonal means. The uncertainty on our regression model output feeds through (has an impact on the total uncertainty on the monthly mean zonal means) to the final

uncertainty but is not solely used (equation 3 of the manuscript). The uncertainties on the regression model fit are expected to be smaller than the uncertainty on the measurements.

The bootstrapping is performed on the monthly mean zonal mean of the difference fields (not the actual measurements or the monthly means thereof).

I also got a bit confused as to what was being bootstrapped. The fit, and thus the residual, is to the monthly mean zonal mean difference field. Is the measurement uncertainty in bootstrap Step 3 that of an individual profile, or of a particular monthly mean zonal mean bin? If the latter (which I was thinking it should be), how is this uncertainty computed?

Yes, we perform the bootstrapping method to obtain an uncertainty estimate on our regression model fit to the monthly mean zonal means of the difference field. The measurement used in the bootstrapping is defined in equation 7 of the manuscript.

At step 4 it seems to go back to the individual profile, but still the residual uncertainties being added are for monthly mean zonal mean values. If it truly is going back to the original data, the new data with the added uncertainties are then re-averaged into monthly mean zonal means, thus beating down some of that variability. Or is it just that each monthly mean zonal mean value is given a random residual to construct the new difference field, and then this new field is fit with the regression model?

That is correct, each monthly mean zonal mean value of the difference field is given a random residual to construct a new difference field which is fitted by the regression model again (this is repeated 200 times).

That approach makes more sense to me, to work with the monthly mean zonal mean values, rather than revert all the way back to individual profiles.

And that is what we are doing.

It seems the month to month noise from the residual is being used to replicate the day to day noise within the month of the individual measurements.

We have updated the text to clarify that we are bootstrapping the monthly mean zonal mean difference fields.

P11 L20: In reference to Figure 3, it would be helpful to state what the original standard was at this level. You can see from the plot that sonde data didn't change, but adding it to the text would be good.

That is a good suggestion from the reviewer and we have stated the standard for that particular level in the revises manuscript.

Also, why are the sonde data so limited before 1995?

In this particular latitude band (25°N to 30°N) not many ozonesonde stations are located. We were able to obtain data from five stations. From those only two started operation before 1995. One station (New Delhi, India) has sporadic measurements during the 1970s and 1980s, and more regular measurements from 1994 onward. The second station (Naha, India) started regular measurements in 1989. However, we only calculate a monthly mean zonal mean if we have at least six measurements available for the respective latitude bin and level (as mentioned in the manuscript). Measurement frequency at ozone sounding stations can vary quite a bit, but on average there might be one sounding a week per station. At the station Naha this is the case, so that only in the mid-1990s enough individual soundings per months were available to actually calculate a monthly mean zonal mean.
For different latitude bands (e.g. further north) the number of available sounding stations and therefore also measurements is different, which means more monthly mean zonal mean values based only on ozonesonde measurements might be available.

How were the SAGE adjusted in this case, is the CTM used over a multi-year period to make matches between SAGE and sonde?

Yes, CTM data are used over a multi-year period. The correction of the satellite data to the ozonesondes (at this level) is done globally, and not by latitude zone. That's why we include Legendre polynomials when calculating the difference fields so that in case there are no ozonesonde measurements in some zone available, the differences in neighbouring zones propagate into that zone so that we can still corrected the, e.g. SAGE data in the zone where only a few ozonesonde measurements are available.

Finally, in reference to Figure 3, is there published literature to compare the stated offsets for MLS and HALOE, or do direct comparisons between HALOE and sonde/MLS and sonde back up these values? Even with diminished match-ups, a difference approaching 40% should be detectable.

There are analyses about differences between ozone soundings and different satellite measurements for different pressure levels published in Davis et al. (2016), Figure 4. The differences that they present are calculated between matched satellite and ozonesonde observations, and therefore are not directly comparable to the differences obtained within our study (and which are shown in Figure 3). However, they also showed very high values for the comparison between HALOE and ozone soundings (up to 55% difference), especially for pressure levels > 80 hPa.
(reference: Davis, S.M., Rosenlof, K.H., Hassler, B., Hurst, D.F., Read, W.G., Vömel, H., Selkirk, H., Fujiwara, M., and Damadeo, R.: The Stratospheric Water and Ozone Satellite Homogenized (SWOOSH) database: a long-term database for climate studies, Earth Syst. Sci. Data, 8, 461–490, doi:10.5194/essd-8-461-2016, 2016)

P15, Eqn 9: Did the authors do any sensitivity testing with the regression fits to two levels, in particular in cases where the ozone at the two levels is highly correlated. I would think that often using fits to both levels is not needed, are there specific situations were the fit to two successive layers is particularly useful?

The reviewer is correct that using both the *N-2th* and *N-1th* levels as predictors for the ozone at level *N* could be unnecessary since the ozone at the two lower levels could be spatially similar. We did conduct tests and the value of using two levels to estimate ozone at the level above is that if there is a horizontally spatially non-uniform trend in the vertical distribution of ozone, as is often the case, then using both the *N-2th* and *N-1th* levels as predictors for the ozone at level N does a much better job than just scaling the ozone at level *N-1*. Because the number of data is far larger than the number of degrees of freedom in a fit that uses both of the levels below to estimate the ozone at level *N*, and because this resulted in better estimates, we decided to use this two-level estimator approach.

P18: The details in the differences in Tier 1.x are difficult to see. Maybe a particular year or event period can be highlighted, showing the full time series for Tier 0 and Tier 0.5, but a shorter sample period for the other data sets to point out specific features.

We changed Figures 6 and 7 (following a suggestion from Reviewer #1) and are now plotting the differences of the Tier 1.x to Tier 0.5 data to highlight the differences between the data sets.

Did the authors look at the consistency between the integrated Tier 0 and Tier 0.5 and the total ozone?

We did compare the integrated Tier 0.5, Tier 1.4 and the total column ozone (TCO) data set that was used to create Tier 0.5. Tier 0 has data gaps in all latitude bands which makes the integration of the full Tier 0 data set not comparable with the TCO data set or integrated Tier 0.5.

I think such a comparison would be useful as a validation point and to show the influence of using the total ozone to fill the data set. With a full profile data set and independent total ozone merged data set from the same group, users might naturally work with both simultaneously, so analyzing their consistency would be useful. This is a validation that the other merged data sets cannot easily do because they do not have full vertical resolution.

The comparison between the integrated Tier 0.5 and Tier 1.4 data sets, and the TCO data set showed a very good agreement (within about 10%) between the three data sets in most latitude bands. Mostly the very high and very low values of the TCO data set are slightly under- or overestimated, respectively, by the Tier 0.5 and Tier 1.4 data sets, as you would expect from data sets based on regression model output. The differences between TCO and Tier 0.5 and Tier 1.4 are larger in the latitude bands 70°-85°S, at the edge and within the region of the ozone hole. The interannual variability in these regions cannot be described well with the basis functions used in the regression for creating the Tier 1.x data sets, and can also not be well represented with the gap-filling method used to create Tier 0.5.

Technical Corrections: P2 L20 monhtly → monthly

Corrected.

P3 L4 beside → besides (or in addition too)

Corrected.

P3 L7 suggest wording change, maybe still have limited coverage in the troposphere

Done.

P3 L11-12 suggest wording change, maybe "This is particularly important when seeking to detect the small but expected signal of ozone recovery due to reductions in ozone depleting substances."

We followed the suggestion by the reviewer and implemented the wording change.

We could not find a 'where' that should be replaced by a 'were' on P3 L24. We assume the reviewer was referring to the sentence '(1) where updated versions of the ozone data sources were available, these were used (Sect. 2)' (P3 L22-23). However, the 'where' in this sentence seems correct to our knowledge.

Corrected.

Done.

Done.

Corrected.

P8 L3 It would be helpful to include the approximate pressure corresponding to 15km here.

Done.

P8 L6 I'm not sure of the meaning of the sentence starting "As each ozonesonde is individually prepared..." I would argue that the inherent physics of the satellite measurements vs sonde is the primary reason the ozonesonde measurements are more reliable in the troposphere (not better calibration/validation). Or are you saying that we know sonde data are more reliable because of the very precise calibration/validation?

Correct, that is what we meant.

In any case I think you could easily remove this sentence. It seems to repeat the sentence before, which stands on its own.

Good suggestion and we have removed that sentence.

P8 L12 '...,or temporally and spatially highly-resolved output from a CTM, ..."

Done.

P9L21 were corrected to the extent possible (remove "for")

Done.

P11 L2: suggest "This step represents the influence of measurement uncertainty on the residual."

Done.

P15 L24 remove "used"

Done.

P19 L1 and Figure 8 Caption: green -> red

Corrected and throughout the text.

P20 L3 (85deg S to 90 deg S, 58 hPa)

Done.

P22 L1 remove "different" or "selected"

Done.

P22 L12 different data sources

Done.

P22 L16 remove "of"

Done.

P23 L4 though some discrepancies

Done.

P23 L23 each individual measurement

Done.

P23 L29 and therefore reduce

We assume the reviewer referred to the sentence 'Additional measurements in the troposphere would help to better constrain the regression model used to fill the data-gaps and therefore reducing the overall uncertainties on the monthly mean zonal mean ozone values.' (L31-33). We corrected the word 'reducing' in this sentence.

P24 L12 remove "planned" (used later in sentence)

Done.

P24 L14 remove "," after measurements

Done.

---

## Author Comment (AC2) · 31 Jul 2018

We wanted to point out that we changed four figures for the revised version of the manuscript. Two figures changed according to reviewer comments (Figure 6 and Figure 7), and two figures (Figure 8 and Figure 9) changed because we realized that we had not used the latest version of the database for the figures in the originally submitted manuscript version. The message of the latter figures did not change (the differences between the older and the most recent version of the database were pretty minor almost everyhwere), therefore we decided to just update the figures and not change the text in the manuscript.

[Figure]

2018.